# Did he or didn't he? Mixed evidence for the continued influence of retracted misinformation on person impressions

Amy J. Mickelberg[1]*, Bradley Walker[1], Ullrich K. H. Ecker[1,2], Piers D. L. Howe[3], Andrew Perfors[3], Nicolas Fay[1]*

**1** School of Psychological Science, University of Western Australia, Crawley, WA, Australia **2** Public Policy Institute, University of Western Australia, Crawley, WA, Australia **3** School of Psychological Sciences, University of Melbourne, Parkville, VIC, Australia

\* amy.mickelberg@research.uwa.edu.au (AJM); nicolas.fay@uwa.edu.au (NF)

## Abstract

Retracted misinformation often continues to influence event-related reasoning, but there is mixed evidence that it influences person impressions. A recent study found no evidence for the continued influence of retracted misinformation on person impressions across four experiments. However, the study used a dynamic impression-rating measure that may have obscured any continued influence effects. Here we report three experiments that tested for the continued influence of retracted misinformation on person impressions using a non-dynamic impression-formation task that is comparable to tasks used in event-related misinformation research. Participants formed an impression of a fictitious person based on a series of behaviour statements. A negative behaviour statement (e.g., "John kicked his pet dog hard in the head when it didn't come when called") was subsequently retracted or not retracted. Evidence for the continued influence of the retracted behaviour statement was found in one experiment; in the other two experiments the retracted misinformation was fully discounted. The mixed findings indicate that, unlike retracted event-related misinformation, retracted person-related misinformation does not consistently show a continued influence effect. Future research should investigate potential moderating factors, such as the attributes of the misinformation and the presence of social-category information about the protagonist, to reveal the mechanisms underlying the continued influence effect in person impressions.

## Introduction

Misinformation has shaped the discourse around many events and contemporary issues, and has been found to negatively affect people's beliefs, judgements, and behaviours [1–5]. One pernicious aspect of misinformation arises from its resistance to correction, with misinformation often continuing to influence people's

**Data availability statement:** All data and analysis files are available from the OSF database: https://osf.io/x3srb/

**Funding:** This research was supported by a Postgraduate Research Scholarship from the Defence Science and Technology Group of the Department of Defence and an Australian Government Research Training Program Scholarship to the first author, an Australian Research Council grant FT190100708 to the third author, and an Office of National Intelligence and Australian Research Council grant NI210100224 to the last author. The funders had no role in study design, data collection and analysis, decision to publish, or preparation of the manuscript.

**Competing interests:** The authors have declared that no competing interests exist.

reasoning after it has been clearly and credibly retracted [for a review see [1], for meta-analyses see [6,7]. There is also some evidence that misinformation (when not retracted) in the form of false allegations can sway people's opinions and impressions of others [8–10]. However, evidence for the continued influence of retracted misinformation on person impressions is mixed, with some studies showing evidence for continued influence [e.g., 11,12] and others not [e.g., 10,13].

One explanation for the mixed results relates to misinformation coherence: retracted misinformation may only continue to influence person impressions when it coheres with other information provided about the person being evaluated. Such coherence tends to be present when misinformation relates to an event [e.g., the misinformation provides a causal explanation of an event; see [14], but is often absent from impression-formation tasks, which tend to present participants with a series of conceptually unrelated behaviour descriptions, on the basis of which participants form a person impression [e.g., 8,13,15]. Mickelberg, Walker, Ecker, Howe et al. [10] adapted the impression-formation task used by Ecker and Rodricks [13] to include misinformation coherence by adding a single coherence-building statement that was congruent or causally related to the misinformation [e.g., misinformation: "... kicked his dog hard in the head"; coherence-building statement: "...took the dog to the vet to treat its head injury"; see also [12]; they found no evidence for the continued influence of retracted misinformation on person impressions. However, unlike other misinformation (or impression-formation) research, Mickelberg, Walker, Ecker, Howe et al. used a dynamic impression-rating measure, in which judgements were made repeatedly over the course of the task. It is possible the dynamic task obscured the continued influence effect by focusing participants' attention on the more recent information [i.e., the retraction; see 16,17]. To assess this possibility, the present study used the same impression-formation task as Mickelberg, Walker, Ecker, Howe et al., but used a non-dynamic measure that is in keeping with other misinformation research [12,13, see also 14,18,19].

A large body of empirical evidence shows that retractions can mitigate, but rarely eliminate, the influence of misinformation, a phenomenon known as the continued influence effect [CIE; 14,20, for reviews see 1,21]. Typically, the CIE is investigated using event-related misinformation. For example, participants are presented with a narrative about a fictitious event (e.g., a warehouse fire) that includes misinformation that is causally related to the event (e.g., negligent storage of flammable materials led to the fire). When the misinformation is later retracted, participants are often found to continue to rely on the misinformation in their inferential reasoning (e.g., mentioning the negligent storage of flammable materials when asked why the fire spread so quickly). This aligns with the broader literature on belief perseverance, which shows that original beliefs tend to persist despite evidence to the contrary [22–25].

One cognitive explanation of the CIE proposes that failures in information integration and memory updating drive the CIE. This updating account posits that people construct a mental event model that is continually updated when new information becomes available [26]. However, retraction of a central piece of information (e.g.,

the likely cause of an event) leaves the model incomplete, that is, a 'gap' is left in the mental model. As such gaps cause psychological discomfort [27], people can give preference to an incorrect but complete mental model (containing the retracted misinformation) rather than a correct but incomplete one (without the retracted misinformation), and therefore continue to rely on the retracted misinformation [14,28]. Support for this account comes from the finding that providing an alternative explanation for an event (e.g., arson caused the warehouse fire) can help people update their mental model, effectively filling the model gap and eliminating the continued influence effect [e.g., 29,30].

Early research has shown that first impressions are robust and resistant to correction in the face of new and incongruent information [22,23,31,32]. In a study using misinformation about (fictional) political candidates, Thorson [12] found that, relative to a no-misinformation control condition, candidates were evaluated more negatively when it was (falsely) alleged they had accepted donations from a convicted felon, even after the allegation was corrected [see also 33,34]. While this effect was observed only when the politician was from the political party opposite to the political allegiance of the participant [i.e., a worldview effect; see 1,35, for discussion], it supports the presence of a CIE in person impressions. These findings are consistent with Mickelberg, Walker, Ecker, and Fay [11], who used misinformation about a patient's mental health diagnosis (i.e., schizophrenia or major depressive disorder) and found that the diagnostic label continued to negatively influence person judgements after it was clearly retracted.

Other studies have failed to detect a CIE when the false allegations are directed at an unfamiliar person who is not tied to a particular group [9,10,13]. In a study by Ecker and Rodricks [13], participants read descriptions of behaviours about a fictitious university student. These behaviours were mostly neutral in valence (e.g., he played a game of social tennis) but some conditions included a critical piece of misinformation (i.e., that he slapped his girlfriend during an argument), which was subsequently retracted or not retracted. Person impressions were recorded at the conclusion of the task (i.e., after the presentation of the entire set of behaviour statements). Here, participants fully updated their impression of the person when the misinformation was corrected, leaving no evidence of a continued influence effect. This suggests that person-related misinformation is more amenable to correction than event-related misinformation [13, see also 36,37].

These mixed findings have prompted further investigation. As briefly mentioned earlier, one factor that may explain the mixed findings is misinformation coherence: the connection between the misinformation and other information provided. In event-related reasoning, it is theorised that misinformation that is causally linked to other information will be more resistant to retraction, as removing it threatens model completeness; typically, the event misinformation used in the continued influence literature is of that nature [14,28,38]. Misinformation coherence may also occur when the misinformation aligns with existing knowledge and beliefs. In the Thorson [12] study, the misinformation that a politician from an opposed political party accepted fraudulent donations may have cohered with participants' prior beliefs (e.g., that politicians from the opposite political party are untrustworthy), making it harder to retract due to high misinformation coherence. Furthermore, misinformation coherence may also explain why Mickelberg, Walker, Ecker, and Fay [11] found that a diagnostic label continued to influence person impressions following its retraction, as the misinformation (i.e., an incorrect mental health diagnosis) cohered with pre-existing beliefs (i.e., mental health stigma). In contrast, in the study by Ecker and Rodricks [13], the misinformation—that a university student slapped his girlfriend—is unlikely to cohere with participants' prior beliefs about typical university students, and did not cohere with the other (neutral) behaviour statements provided.

Mickelberg, Walker, Ecker, Howe et al. [10] extended Ecker and Rodricks' [13] study by including misinformation coherence. Participants received a statement that provided negative misinformation about a person's behaviour (e.g., "John had an affair with his best friend's wife") that was later retracted (or not retracted). In addition, coherence-building information was presented that was congruent with (e.g., "John cheated in a card game") or causally related to the misinformation (e.g., "John was spotted in a hotel lobby with his best friend's wife"). Despite including coherence-building elements, no evidence of a CIE in person impressions was found.

The results of Mickelberg, Walker, Ecker, Howe et al. [10] suggest that misinformation coherence may not be critical to a CIE in person impressions, and by extension, that the CIE may not reliably occur in the context of forming person

impressions. However, the null findings reported may have resulted from the experimental method adopted; Mickelberg, Walker, Ecker, Howe et al. used a dynamic impression-rating task, which required participants to update and record their person impressions repeatedly throughout the task. This contrasts with the traditional CIE paradigm, in which judgements are measured once, at the end of the task [11–14]. The mode of responding (i.e., dynamic vs. non-dynamic) may influence how information is updated in memory. In the context of impression formation, this has been conceptualised in the belief-adjustment model [16, see also 17], and supported by empirical evidence. For example, Stewart [39] found that when impression ratings were recorded once at the end of the task, a primacy effect was observed (i.e., earlier information affected impressions more), whereas when impression ratings were recorded repeatedly, after each item was presented, a recency effect was observed [i.e., later information affected impressions more; see also [40,41]. This suggests that Mickelberg, Walker, Ecker, Howe et al.'s dynamic measure may have led participants to place greater weight on the retraction than on the misinformation presented earlier when rating their person impression. If correct, this would explain why the retracted misinformation was fully discounted in that study. It also implies that a CIE in person impressions could be promoted by a non-dynamic end-of-task impression rating measure, placing more focus on the misinformation than on the later retraction. This was tested in the present study.

Across three experiments, we tested for a continued influence effect (CIE) using a non-dynamic impression-formation task that included a misinformation statement that cohered with other person-related information presented to participants. Following Mickelberg, Walker, Ecker, Howe et al. [10], we presented participants with 27 behaviour statements that related to a person named "John". These mostly comprised neutral statements (e.g., "John likes to sing his favorite song in the car") and a negative target misinformation statement (e.g., "John had an affair with his best friend's wife") that was or was not subsequently retracted. To build misinformation coherence, a statement closely aligned to the misinformation (a coherence-building statement, e.g., "John and his wife met with a marriage therapist") was also included. Experiments 1, 2, and 3 used different target misinformation statements and different coherence-building behaviour statements. This was done to assess the robustness of any continued influence effect observed.

Each experiment included three conditions. In the negative retraction condition, participants were presented with a negative target statement about the protagonist that was later retracted. In the negative no-retraction condition the same negative target statement was presented but was not retracted. In the neutral condition, the negative target statement was replaced by a neutral target statement about the protagonist; this condition served as a no-misinformation control condition. We hypothesised that compared to the control condition, misinformation (before retraction) would negatively affect person impressions. Our key prediction was that, consistent with a CIE in person impressions, the retracted misinformation would continue to influence participants' impression of the target.

## Experiment 1

Experiment 1 used statements from Mickelberg, Walker, Ecker, Howe et al. [[10]; Experiment 3] to assess whether a CIE in person impressions emerges when using coherence-building elements and a non-dynamic rating task.

### Method

**Participants.** A-priori power analysis [using G*Power 3; 42] suggested a minimum sample size of 244 participants to detect a small-to-medium effect, $f^2$ =.04, at α =.05, 1- β =.80. A sample of 301 U.S.-based Prolific workers were recruited through the online crowd-sourcing platform Prolific (https://prolific.com). Participants (152 women, 143 men, 5 other gender, 1 preferred not to say) were aged 18–78 years (M = 38.83, SD = 13.92). Participants were randomly assigned to one of the three conditions (neutral, negative retraction, negative no-retraction), with the constraint of approximately equal cell sizes (n = 100). All participants were unique (i.e., each participant was able to participate in only one of the three experiments); participants of Mickelberg, Walker, Ecker, Howe et al. [10] were excluded from participation.

**Impression-formation task.** The impression-formation task was adapted from Mickelberg, Walker, Ecker, Howe et al. [10]. Participants were told they should form an impression of "John" and that four of John's acquaintances had provided examples of his behaviour that they had observed over the past few months. The 27 behaviour statements were then presented to participants one by one, in two phases: (1) presentation of initial information (trials 1–17), and (2) presentation of updated information (trials 18–27; see Fig 1). The first phase included 15 neutral filler statements, presented in a random order, the target statement presented at trial 13, and a coherence-building statement presented at trial 15. Target-statement valence depended on the condition (negative vs. neutral). All conditions received the coherence-building statement (note this only served to build coherence in the conditions featuring the misinformation target statement). The second phase presented 10 updates; these included six additional neutral filler statements not presented in the first phase (e.g., "ADDITION: John did really well at the quiz night"), two confirmations of neutral filler statements from the first phase (e.g., "CONFIRMATION: John was running late, so he drove to work rather than taking the bus"),

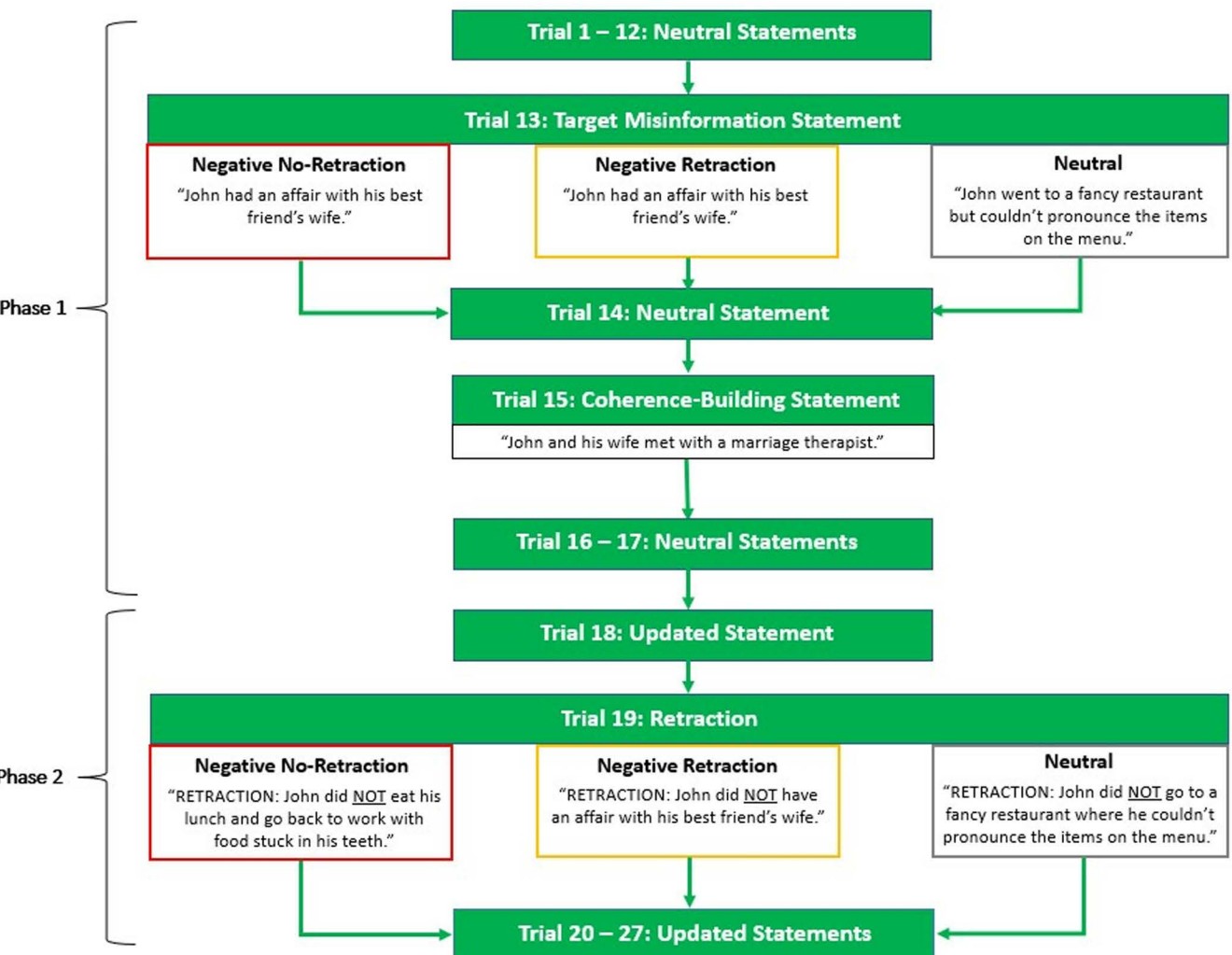

**Fig 1. Impression-formation task.** Neutral statements referred to generic behaviours of the protagonist John. The target statement was presented at trial 13; in the negative retraction and neutral conditions, it was retracted at trial 19. The coherence-building statement was presented at trial 15.

 

and two retractions of statements from the first phase. In the negative retraction and neutral conditions, this included the retraction of the respective target statement (e.g., "RETRACTION: John did <u>NOT</u> have an affair with his best friend's wife"), as well as a retraction of a neutral filler statement (e.g., "RETRACTION: John was <u>NOT</u> singing loudly to his favorite song in the car") In the negative no-retraction condition, two neutral filler statements were retracted. The update statements were presented in a random order, except that the target-statement retraction always occurred at trial 19.

**Behavioural stimuli.** The neutral behaviour statements and the target behaviour statement were selected from a corpus of stimuli that were pre-rated on morality (from −4, *very morally bad*, to +4, *very morally good*) and believability [from 0, *not believable*, to 8, *very believable*; [43]]. The neutral filler and target statements were selected to be neutral in morality (≈ 0) and high in believability (> 6); e.g., the neutral target statement "John went to a fancy restaurant but couldn't pronounce the items on the menu" was associated with morality = 0.06, believability = 6.75. The negative target statement was selected to be strongly immoral and high in believability. In Experiment 1, the statement was "John had an affair with his best friend's wife" (morality = −3.40, believability = 6.84).

The coherence-building statement was selected from a pilot test. Seventeen such statements (9 and 8 statements relating to the negative target statements of Experiments 1 and 2, respectively) were created. For the pilot test results regarding the coherence-building statement in Experiment 2, refer to the Experiment 2 Method section. In the pilot test, $N$ = 100 Prolific participants rated statements on morality and believability [following 43], and also rated how likely the coherence-building statement was to imply the negative target behaviour (in absence of the associated target statement), and how well the coherence-building statement explained the target behaviour (in the presence of the associated target statement), on scales of 1 (*extremely unlikely*) to 7 (*extremely likely*). The coherence-building statement most neutral in morality (≈ 0), high in believability (> 7), unlikely to imply the target statement (< 4), but likely to explain the target statement (> 4) was selected. These criteria were chosen because the coherence-building statement was required to be neutral so as not to have a direct influence on person impressions, and highly believable so as not to be ignored. In addition, the coherence-building statement should not independently imply the target statement in its absence (low 'imply' rating) as this would interfere with the neutrality of the control condition, but should provide a logical explanation for when the target statement was presented (high 'explain' rating). The coherence-building statement selected was "John and his wife met with a marriage therapist" (morality = 2.14, believability = 7.16, imply = 3.48, explain = 4.94). The causally-related statement used in Mickelberg, Ecker, Walker, Howe et al. [[10]; Experiment 3] was included in the pilot test but did not meet the current selection criteria and was therefore not selected for the present study. See S2 File for the full list of behaviour statements selected across all experiments.

**Reysen likability scale.** John's likability was measured with the Reysen Likability Scale [44]. The scale consists of 11 items (e.g., "This person is warm"), each rated on a Likert scale from 1 (*very strongly disagree*) to 7 (*very strongly agree*; see S2 File for the full scale). The scale was previously used by Ecker and Rodricks [13] and Mickelberg, Walker, Ecker, Howe et al. [10] to test for a CIE in person impressions, and has been associated with strong internal consistency reliability [Cronbach's α =.88; [44]]. Across Experiments 1–3, reliability ranged from α =.89 to α =.94.

**Inference questions.** Seven inference questions were used to measure misinformation reliance during inferential reasoning (two open-ended questions and five Likert rating scales). The two open-ended questions were presented as the first and last items and asked participants for information about John ("If you could tell someone about one specific thing John has done, what would it be?"; "Describe briefly in one sentence what kind of relationship John has with his best friend's wife"). The other five inference questions asked participants to what extent they endorsed statements about John (e.g., "John should be ashamed of his behaviour"; see S2 File) and participants responded on an 11-point Likert scale from 0 (*strongly disagree*) to 10 (*strongly agree*). In Experiment 1, after the inference questions, an explicit target-behaviour question was added, asking participants how likely it was that John was having an affair (0 [extremely unlikely] to 10 [extremely likely]). This was done to determine if explicit references to the misinformation would affect participants' reliance on the misinformation. This was not the case (see S3 File).

**Recognition test.** A recognition test was included to measure the extent to which participants attended to and encoded the behaviour statements. Eleven multiple-choice questions, each with three to four response options, tested recognition of the initial and updated behaviour statements (e.g., "What language is John learning?"—"Russian", "Japanese", "French"; see S2 File for the full list of questions).

**Procedure.** The experiments were conducted online. Participants received an information sheet approved by the University of Western Australia's Human Research Ethics Office and provided informed consent (by clicking 'next' to proceed to the study). After providing informed consent, participants completed the impression-formation task, followed by the likability scale, inference questions, and recognition test, and were then debriefed. Median completion time was approximately 8 min. Participants were paid £1.00 (approximately US$1.25).

## Results

The data were analysed using linear regression models. All analyses were performed and all figures were created in R [v4.0.3; 45]. Regression models were estimated using the *lm* function of base R. Figures were created using *ggplot2* [v3.3.5; 46]. The data, R script, and supplementary information associated with all experiments are available at https://osf.io/x3srb/. Performance on the recognition test was high ($M$ = 88.46%, $SD$ = 13.76%). Nine participants scored below 50%, but excluding them did not qualitatively affect the results, so the full sample was used for the analyses.

**Likability scores.** Likability scores were entered into a linear regression with condition as a predictor (negative retraction; negative no-retraction; neutral). The model was significant, $F(2, 298)$ = 23.32, $p$ <.001, $R^2$ =.14. Likability was lower in the negative no-retraction condition compared to the neutral control condition, β = −0.74, $SE$ = 0.12, $t$ = −6.30, $p$ <.001, $f^2$ = 0.20, and compared to the negative retraction condition, β = −0.58, $SE$ = 0.12, $t$ = −5.04, $p$ <.001, $f^2$ = 0.13, indicating the negative target statement was effective in lowering participants' impressions of the protagonist when not retracted. There was no statistical evidence of a difference between the negative retraction condition and the neutral control condition, β = −0.15, $SE$ = 0.11, $t$ = −1.41, $p$ =.161, $f^2$ < 0.01, suggesting the retraction was fully effective (i.e., no CIE; see Fig 2).

**Inference scores.** Open-ended inference responses were scored by two naïve scorers following a standardised guide specific to each experiment (see S2 File). Any scoring discrepancies were resolved through discussion. Unambiguous references to the target statement were scored 1 (e.g., "John did have an affair"). References to the misinformation suggesting an ambiguous level of endorsement were scored 0.5 (e.g., "John may have had an affair?"). Responses were scored 0 when the misinformation was not mentioned or was discredited (e.g., "It was suggested that John had an affair but that was later retracted"). To put the Likert scale inference questions on the same scale as the open-ended questions, responses were divided by 10; scales that were positively worded were reverse-scored. All responses were then summed to create an overall inference score, which ranged from 0 to 7, with higher scores indicating stronger references to the misinformation.

Inference scores were moderately negatively correlated with likability ratings ($r$ = −.58) but not strongly enough to be considered collinear. Inference scores were entered into a linear regression with condition as a predictor (negative retraction; negative no-retraction; neutral). The model was significant, $F(2, 298)$ = 323.90, $p$ <.001, $R^2$ =.68. Inference scores were higher (i.e., stronger reliance on the misinformation) in the negative no-retraction condition compared to the neutral control condition, β = 3.69, $SE$ = 0.17, $t$ = 21.73, $p$ <.001, $f^2$ = 2.37, and compared to the negative retraction condition, β = 3.46, $SE$ = 0.18, $t$ = 18.89, $p$ <.001, $f^2$ = 1.79, indicating there was an effect of the negative target statement on inferential reasoning when not retracted. There was no statistical evidence of a difference between the negative retraction and neutral conditions, β = 0.23, $SE$ = 0.13, $t$ = 1.74, $p$ =.084, $f^2$ = 0.02. This suggests the retraction was fully effective and there was no evidence of a CIE in inferential reasoning (see Fig 3).

**Equivalence analysis.** There was no statistical evidence for a CIE in Experiment 1. To determine whether there was support for the null hypothesis (i.e., statistical evidence against the CIE in person impressions), an equivalence analysis

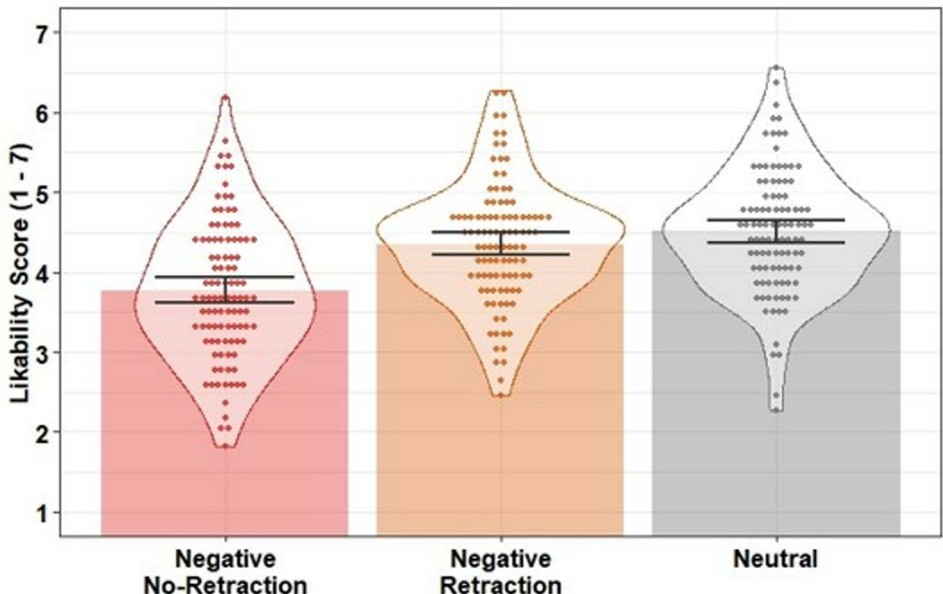

**Fig 2. Mean likability scores across conditions in Experiment 1.** Bars indicate condition means; error bars are 95% bootstrapped confidence intervals (1000 resamples); data points indicate participant mean scores; violins provide distributional information.

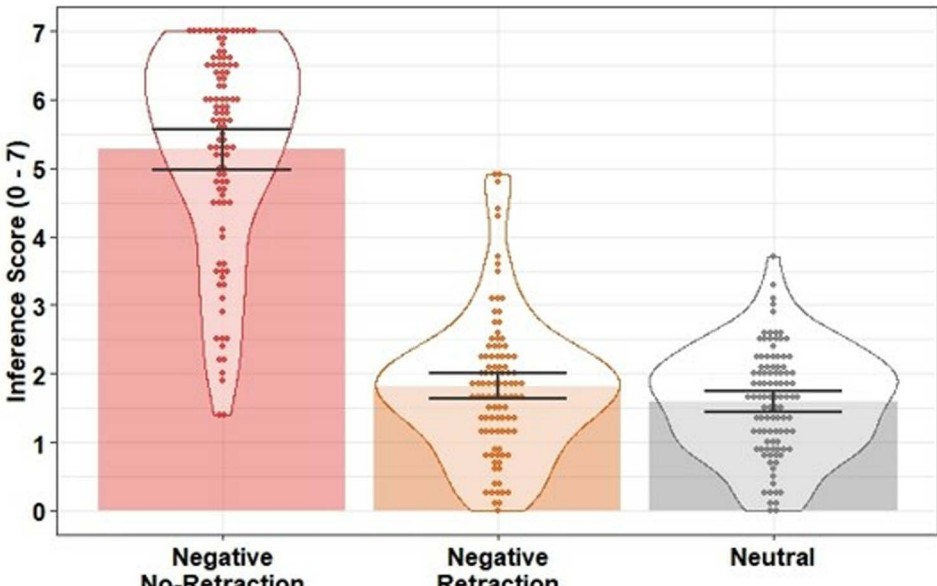

**Fig 3. Mean inference scores across conditions in Experiment 1.** Bars indicate condition means; error bars are 95% bootstrapped confidence intervals (1000 resamples); data points indicate participant mean scores; violins provide distributional information.

was conducted on the non-significant linear regression results between the negative-retraction and the neutral condition in Experiment 1, using likability and inference scores. The Anderson-Hauck procedure [47] was employed using the *reg.equiv* function in R [48]. Standardized regression coefficients of ±0.1 were chosen as the equivalence bounds, with anything falling within this range considered a negligible effect. The equivalence analysis for Experiment 1 indicated insufficient evidence for negligible effects across all measures (likability and inference scores). Thus, while there was an absence of evidence for a CIE in person impressions, there was no conclusive evidence for or against the CIE in person impressions.

## Discussion

Experiment 1 tested whether the continued influence effect is observed in person impressions when misinformation is accompanied by coherence-building elements and impression ratings are recorded using a non-dynamic rating measure. Results showed that negative misinformation that was not retracted had a negative impact on person impressions, and that upon its retraction, any influence of the misinformation was fully discounted (i.e., there was no CIE). This was found for both likability [as in [13]] and inferential reasoning, replicating the findings of Mickelberg, Walker, Ecker, Howe et al. [10]. Given there was no evidence of a CIE in person impressions, and the equivalence analysis returned inconclusive findings for or against the null hypothesis, Experiment 2 provided a direct replication with different target materials.

## Experiment 2

Experiment 2 tested whether the inclusion of a different target statement and a different coherence-building statement [from [10]; Experiment 4] would promote a CIE in person impressions when using coherence-building elements and a non-dynamic rating measure of person impressions.

### Method

**Participants.** A sample of 300 U.S.-based Prolific workers (199 women, 94 men, 6 other gender, 1 preferred not to say) aged 18–78 (*M* = 36.12, *SD* = 12.83) were recruited. Participants were randomly assigned to one of three conditions (neutral, negative retraction, negative no-retraction), with the constraint of approximately equal cell sizes.

**Impression-formation task.** The task in Experiment 2 was identical to that used in Experiment 1, apart from the different target and coherence-building statements. In addition, the explicit target-behaviour question used in Experiment 1 was replaced with a final, overall impression rating. Participants were asked to rate their overall impression of John on a scale from −50 (extremely positive) to 50 (extremely negative). The results of the final impression rating can be found in the S3 File.

The target statement followed Mickelberg, Walker, Ecker, Howe et al. [[10]; Experiment 4]. The target statement was "John kicked his pet dog hard in the head when it didn't come when called" (morality = −3.55, believability = 6.79). The coherence-building statement was selected from the same pilot test as per Experiment 1. The selected statement was "John accompanied his wife to the vet to have his dog treated for a head injury". The pilot test showed it to be mostly neutral in morality (morality = 2.65), high in believability (believability = 7.20), unlikely to imply the target statement (imply = 3.21), but likely to explain the target statement (explain = 4.40; see S1 File for details). The causally-related statement from Mickelberg et al. [[10]; Experiment 4] was included in the pilot testing. Pilot test results showed it met the current selection criteria and it was therefore selected for the present study.

**Inference questions.** The inference questions and scoring guide were based on those used in Experiment 1, but were updated to reflect the misinformation statement used in Experiment 2 (e.g., "…how do you think John's pet dog could have received a head injury?"; see S2 File).

**Procedure.** The procedure was identical to that in Experiment 1. The median completion time was approximately 8 min, and participants were paid £1.20 (approximately US$1.50).

## Results

Performance on the recognition test was high (*M* = 91.68%, *SD* = 8.41%). No participant scored below 50%.

### Likability scores

Likability scores were entered into a linear regression with condition as a predictor (negative retraction; negative no-retraction; neutral). The model was significant, $F(2, 297) = 95.06$, $p < .001$, $R^2 = .39$. Likability was lower in the negative no-retraction condition compared to the neutral control condition, $\beta = -1.51$, $SE = 0.12$, $t = -12.73$, $p < .001$, $f^2 = 0.81$, and the negative retraction condition, $\beta = -1.21$, $SE = 0.13$, $t = -9.61$, $p < .001$, $f^2 = 0.47$, indicating the negative target statement was effective in lowering participants' impression of the protagonist when not retracted. In contrast to Experiment 1, there was statistical evidence of a difference between the negative retraction and neutral control conditions, $\beta = -0.31$, $SE = 0.10$, $t = -2.96$, $p = .003$, $f^2 = 0.04$, suggesting that the retraction was not fully effective and the retracted misinformation continued to influence person impressions [with a small effect size according to [49]; see Fig 4].

### Inference scores

Inference scores were strongly negatively correlated with likability ratings (*r* = −.80) but were treated as separate measures. Inference scores were entered into a linear regression with condition as a predictor (negative retraction; negative no-retraction; neutral). The model was significant, $F(2, 297) = 468.90$, $p < .001$, $R^2 = .76$. Inference scores were higher (i.e., stronger reliance on the misinformation) in the negative no-retraction condition compared to the neutral control condition, $\beta = 4.22$, $SE = 0.15$, $t = 27.27$, $p < .001$, $f^2 = 3.74$, and compared to the negative retraction condition, $\beta = 3.73$, $SE = 0.17$,

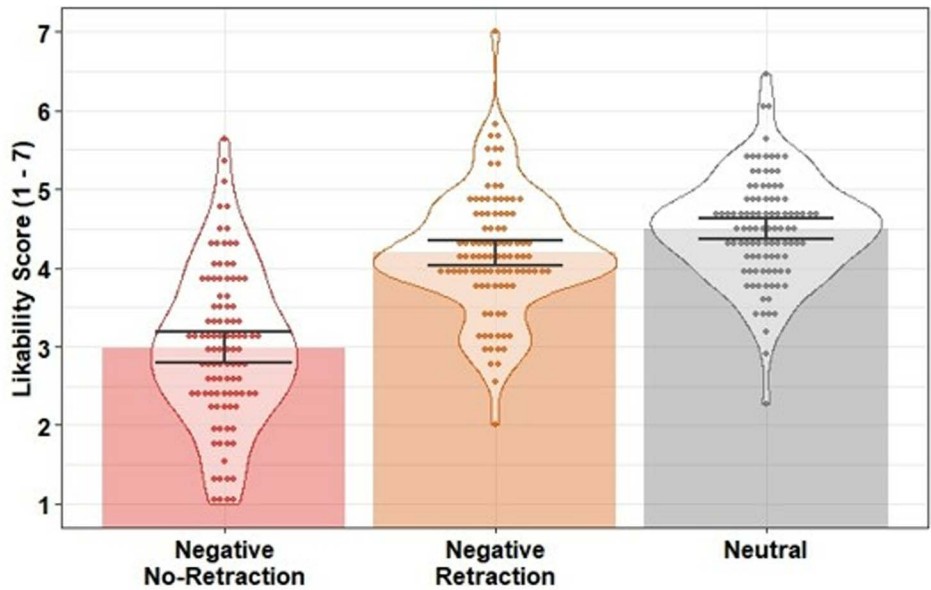

**Fig 4. Mean likability scores across conditions in Experiment 2.** Bars indicate condition means; error bars are 95% bootstrapped confidence intervals (1000 resamples); data points indicate participant mean scores; violins provide distributional information.

$t = 21.68$, $p <.001$, $f^2 = 2.37$. In contrast to Experiment 1, there were higher inference scores in the negative retraction condition than in the neutral control condition, $\beta = 0.49$, $SE = 0.12$, $t = 4.01$, $p <.001$, $f^2 = 0.08$. This suggests that the retraction was not fully effective and there was evidence of a small CIE in inferential reasoning [a small effect size according to [49]; see Fig 5].

## Discussion

The Experiment 2 results suggested a negative impact of (non-retracted) negative misinformation on person impressions, replicating Experiment 1 and Mickelberg, Walker, Ecker, Howe et al. [10]. Once the misinformation was retracted, Experiment 2 showed evidence of a continued influence effect in both likability and inferential-reasoning scores. This pattern of results contrasts with Experiment 1, which showed no evidence of a CIE in person impressions. The evidence of continued influence also contrasts with Mickelberg, Walker, Ecker, Howe et al.'s Experiment 4, which used the same target materials but included a dynamic impression-rating measure. Taken together, this suggests that task design can have an effect on the continued influence in impression formation [see [16],[17]]. That said, and given the inconsistent results between Experiments 1 and 2, this impact may be limited to the specific misinformation statement used. In light of the mixed findings reported across Experiments 1 and 2, a third and final experiment was run with different target materials to test the robustness of the effects observed in Experiment 2.

## Experiment 3

Experiment 3 again tested whether a CIE in person impressions is observed when using both coherence-building elements and a non-dynamic rating measure. As Experiments 1 and 2 provided mixed evidence for a CIE in person impressions, Experiment 3 employed new target materials to clarify the conflicting findings.

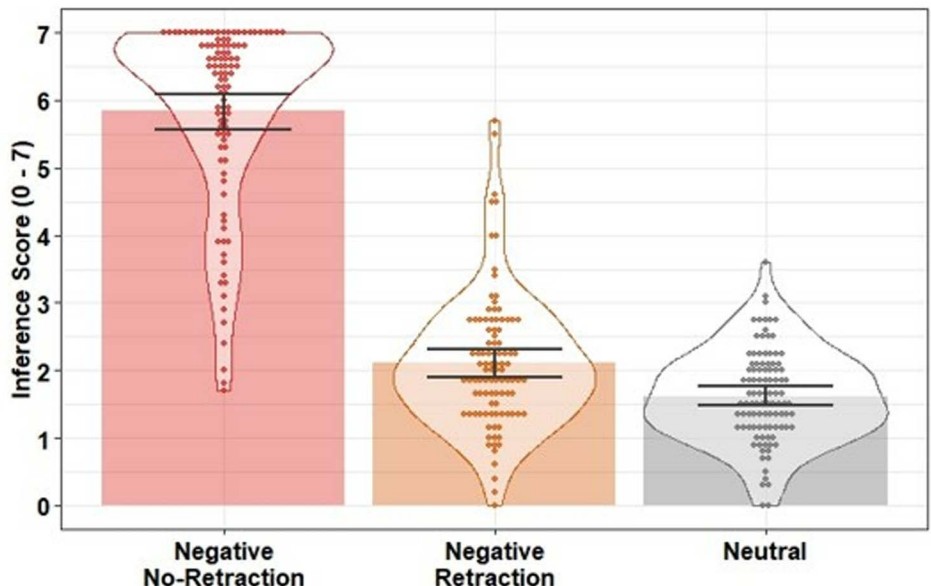

**Fig 5. Mean inference scores across conditions in Experiment 2.** Bars indicate condition means; error bars are 95% bootstrapped confidence intervals (1000 resamples); data points indicate participant mean scores; violins provide distributional information.

## Method

**Participants.** A sample of 301 U.S.-based Prolific workers (152 women, 148 men, 1 other gender) aged 19–85 ($M$ = 41.95, $SD$ = 14.00) were recruited. Participants were randomly assigned to one of the three conditions (neutral, negative retraction, negative no-retraction), with the constraint of approximately equal cell sizes.

**Impression-formation task.** The task in Experiment 3 was identical to that used in Experiments 1 and 2, apart from the different target and coherence-building statements. In addition, a final, overall impression rating measure was included as per Experiment 2 (see S3 File for results). The target statement was generated by the authors to be low in morality and high in believability (as per Experiments 1 and 2). The statement was "John cut down his neighbors' beloved chestnut tree while they were out, as he was fed up with it obstructing his view." Pilot testing showed that it met the requirements (morality = −3.27, believability = 5.11; see S2 File for details). The coherence-building statement was selected from a separate pilot test following the same selection criteria as Experiments 1 and 2. The selected statement was "John enjoys uninterrupted views of the river from his living room now that his neighbors' tree has been removed" (morality = 0.28, believability = 6.78, imply = 3.95, explain = 5.60; see S2 File for details).

**Inference questions.** The inference questions and scoring guide were based on those used in Experiments 1 and 2, but were updated to reflect the misinformation statement used in Experiment 3 (e.g., "…what happened to John's neighbors' chestnut tree?"; see S2 File).

**Procedure.** The procedure was identical to that of Experiments 1 and 2. The median completion time was approximately 8 min, and participants were paid £1.00 (approximately US$1.25).

## Results

Performance on the recognition test was high ($M$ = 89.64%, $SD$ = 11.48%). Five participants scored below 50% but excluding them did not qualitatively affect the results, so the full sample was used for the analyses.

**Likability scores.** Likability scores were entered into a linear regression with condition as a predictor (negative retraction; negative no-retraction; neutral). The model was significant, $F(2, 298) = 25.75$, $p < .001$, $R^2 = .15$. Likability was lower in the negative no-retraction condition compared to the neutral control condition, $\beta = -0.69$, $SE = 0.12$, $t = -5.93$, $p < .001$, $f^2 = 0.18$, and compared to the negative retraction condition, $\beta = -0.76$, $SE = 0.12$, $t = -6.13$, $p < .001$, $f^2 = 0.19$, replicating the negative effect of the non-retracted negative target statement in Experiments 1 and 2. There was no statistical evidence of a difference between the negative retraction and the neutral control condition, $\beta = 0.08$, $SE = 0.11$, $t = 0.65$, $p = .516$, $f^2 < 0.01$, suggesting the retraction was fully effective (i.e., no CIE; see Fig 6).

**Inference scores.** Inference scores were moderately negatively correlated with likability ratings ($r = -.58$) but not strongly enough to be considered collinear. Inference responses were scored as in Experiments 1 and 2; they were then entered into a linear regression with condition as a predictor (negative retraction; negative no-retraction; neutral). The model was significant, $F(2, 298) = 353.60$, $p < .001$, $R^2 = .70$. Inference scores were higher (i.e., stronger reliance on the misinformation) in the negative no-retraction condition compared to the neutral control condition, $\beta = 3.74$, $SE = 0.17$, $t = 21.82$, $p < .001$, $f^2 = 2.38$, and compared to the negative retraction condition, $\beta = 3.71$, $SE = 0.19$, $t = 19.62$, $p < .001$, $f^2 = 1.95$, replicating Experiments 1 and 2. There was no statistical evidence of a difference between the negative retraction and the neutral control condition, $\beta = 0.02$, $SE = 0.12$, $t = 0.21$, $p = .832$, $f^2 < 0.01$. This suggests the retraction was fully effective and there was no evidence of a CIE in inferential reasoning (see Fig 7).

## Equivalence analysis

Experiment 3 results showed no statistical evidence for a CIE in person impressions. To determine whether there was support for the null hypothesis, an equivalence analysis was conducted on the non-significant linear regression results between the negative-retraction and the neutral control conditions in Experiment 3, using likability and inference scores,

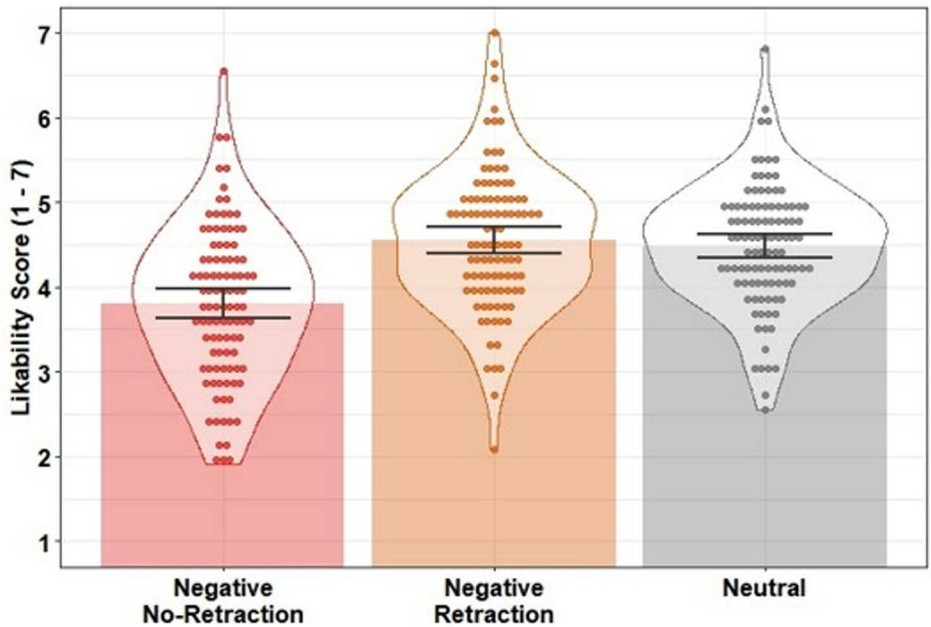

**Fig 6. Mean likability scores across conditions in Experiment 3.** Bars indicate condition means; error bars are 95% bootstrapped confidence intervals (1000 resamples); data points indicate participant mean scores; violins provide distributional information.

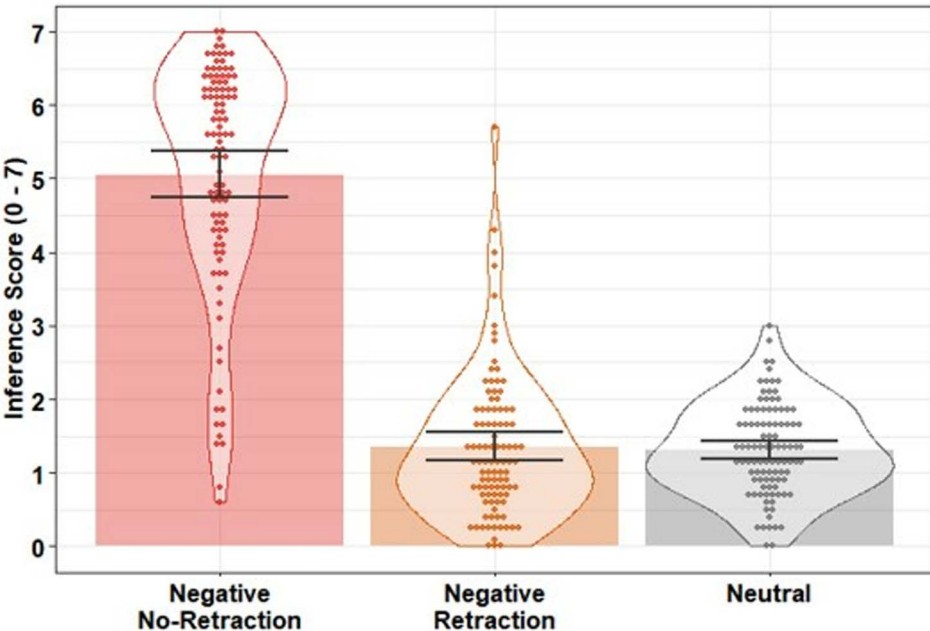

**Fig 7. Mean inference scores across conditions in Experiment 3.** Bars indicate condition means; error bars are 95% bootstrapped confidence intervals (1000 resamples); data points indicate participant mean scores; violins provide distributional information.

following the same procedure as Experiment 1. The equivalence analysis for Experiment 3 indicated insufficient evidence for negligible effects across all measures (likability and inference scores). Thus, while there was an absence of evidence for a CIE in person impressions, there was no conclusive evidence for or against the CIE (see S3 File).

## Discussion

Experiment 3 showed the same pattern of results as Experiment 1 using different target and coherence-building statements. Non-retracted negative misinformation had a negative impact on person impressions, and a clear retraction eliminated the influence of the misinformation (i.e., there was no evidence of a CIE in person impressions). This was found for both likability and inferential reasoning, replicating the null results reported in Experiment 1 and in Mickelberg, Walker, Ecker, Howe et al. [10]. The results of Experiments 1 and 3 contrast with the positive results reported in Experiment 2, which showed evidence of a CIE in person impressions. Overall, the findings from the three experimental studies indicate mixed evidence for the CIE in person impressions.

## Internal meta-analysis

Given the inconsistent findings across the three experiments reported here, we performed an internal meta-analysis to determine the summary effect of continued influence on person impressions [following 50]. The random-effects model was estimated using the *meta* package in R [51] using the inverse-variance method. A random-effects model was chosen as effect sizes varied considerably between studies. This approach incorporates both within-study and between-study variance, leading to a more generalisable estimate of effect size [50,52]. Standardized coefficients were selected as the effect size. Results showed there was no statistical evidence of a summary effect of continued influence on likability ratings, $\beta = -0.13$, $Z = -1.14$, $p = .253$, 95% CI [$-0.35$, $0.09$]. A nonsignificant summary effect was also shown for continued influence on inferential reasoning, $\beta = 0.25$, $Z = 1.79$, $p = .073$, 95% CI [$-0.02$, $0.52$]. These findings indicate that there is no conclusive evidence of a CIE on likability or inferential reasoning when examining the combined results of the three experiments. When also including the experiments from Mickelberg, Walker, Ecker, Howe et al. [10] in the meta-analysis, there was again no conclusive evidence of a CIE on either likability (four experiments; $\beta = -0.07$, $Z = -1.12$, $p = .263$, 95% CI [$-0.18$, $0.05$]) or inferential reasoning (two experiments; $\beta = -0.03$, $Z = -0.11$, $p = .909$, 95% CI [$-0.62$, $0.55$]).

## General discussion

The experiments reported tested if negative misinformation continues to influence person impressions after it is retracted (the continued influence effect, CIE). In Experiments 1–3, the misinformation cohered with the other information provided about the protagonist, and person impressions were assessed using non-dynamic end-of-task measures [rather than being repeatedly measured throughout the task, as per [10]]. To assess the robustness of the CIE in person impressions, each experiment used different negative misinformation and coherence-building information. When the negative misinformation was not retracted, it consistently lowered participants' perception of the protagonist across all experiments, underscoring the impact a single piece of highly immoral negative information can have on shaping person impressions [53,54], for a review see 55]. When the negative misinformation was retracted, Experiments 1 and 3 found no statistical support for a CIE in person impressions. In contrast, a CIE in person impressions was observed in Experiment 2. Given the mixed findings, an internal meta-analysis was conducted to determine the overall trend of a CIE in person impressions. This returned no conclusive evidence for a CIE across the three experiments, and it remained inconclusive when the four experiments of Mickelberg, Walker, Ecker, Howe et al. [10] were also included.

Our mixed results could be due to the true CIE in person impressions being very small. This is compatible with the findings observed in Experiment 1 ($f^2 \approx 0.01$), Experiment 2 ($f^2 \approx 0.06$), and Experiment 3 ($f^2 < 0.01$) and might also account for the mixed pattern of results observed in prior studies. Given the typical sample size used in the field, a small true effect can be expected to produce a mix of null results [Experiments 1 and 3; [9],[10],[13]] and significant results [Experiment 2;

[11],[12],[34]]. While investigation of misinformation reliance and its continued impact on person impressions is of some importance, studying this phenomenon may require greater statistical power than is usually attained in person-impression studies [56].

The mixed results also suggest that the presence of the CIE in person impressions may be dependent on certain conditions. One possibility is that the nature of the task itself can influence the emergence of a CIE in person impressions. Using a dynamic impression-rating task, Mickelberg, Walker, Ecker, Howe et al. [10] found no evidence of a CIE in person impressions across four experiments. However, the dynamic task may have obscured the CIE by focusing participants' attention on the more recent retraction [see [16],[17]]. The CIE observed in Experiment 2 of the present study, which used a non-dynamic task, supports this explanation. Against this, no evidence for a CIE was observed in Experiments 1 and 3 despite the experiments also using a non-dynamic impression-rating task. The nature of the task itself can therefore not satisfactorily explain the observed results.

Alternatively, there may be specific attributes of the target misinformation statement used in Experiment 2 that may have contributed to the emergence of a CIE. While care was taken to select materials that were comparable on core dimensions—the target statements used in Experiments 1–3 had similar morality and believability ratings—we cannot rule out the possibility that a specific characteristic of the Experiment 2 materials promoted the observed CIE. For example, the Experiment 2 target statement related to an extremely violent behaviour (kicking a pet dog) and may have tapped into a different moral dimension than the other target statements [e.g., cruelty vs. dishonesty, see 57,58]. Notably, the Experiment 2 target statement elicited stronger negative person impressions (i.e., lower likability, $M = 2.98$, $SD = 0.98$) than the target statements used in Experiment 1 ($M = 3.77$, $SD = 0.87$) and Experiment 3 ($M = 3.80$, $SD = 0.91$). Future work could explore whether the CIE in person impressions is dependent on misinformation statements that tap into specific moral dimensions [43,59,60] compared to non-moral dimensions [61,62]. For example, certain moral and violent behaviours ("… kicked a dog hard in the head…"; Experiment 2) may activate moral foundations [e.g., care vs. harm; [59]] that potentially lead this behaviour to be perceived as particularly diagnostic of a person's character [8], making a CIE more likely to arise. In contrast, other morally questionable and violent behaviours [e.g., "slapped his girlfriend during an argument"; [13]] may tap into different moral dimensions [e.g., loyalty vs. disloyalty; [59]], potentially explaining the mixed results, although this interpretation is speculative given the two behaviours appear similar at face value. Additionally, misinformation statements that are more extreme often elicit stronger negative reactions [1,63–65], potentially influencing the strength of the CIE in person impressions [see also 66,67]. While a focus on more extreme cases would limit generalizability, it is important for future work to make controlled comparisons of various moral behaviours and the dimensions they tap into [61,68] to conclusively determine their impact on the CIE in person impressions.

While the believability of the misinformation statement was taken into consideration, future work may wish to manipulate the credibility of the retraction, as this may influence the observation of the CIE in person impressions. Research has shown that the continued influence effect can occur when people fail to accept the retraction [69–71], although it should be noted that even with highly credible corrections, the CIE has been shown to occur [see [19]]. Individual differences may also influence the effectiveness of retractions. For instance, individuals with a counterfactual mindset—that is, those with the inclination and ability to spontaneously generate alternative explanations—are more likely to revise their beliefs when presented with contradictory evidence [72]. Future research would benefit from exploring the myriad drivers that may influence a retraction's effectiveness as it relates to person impressions.

The mixed evidence for a CIE in person impressions in the present study contrasts with the robust findings of a CIE in event-related reasoning studies [see [1] for a review, and [6],[7] for meta-analyses]. This suggests that event- and person-related misinformation may be processed differently. Event-related misinformation may be more likely to assert continued influence because events and the associated mental event models tend to have clear cause-and-effect relationships [e.g., a warehouse fire caused by flammable materials, [14]]. Specific event effects also demand a causal explanation (e.g., a fire must be caused by *something*). In this case, a retraction that undermines a causal explanation may be

disregarded to maintain a coherent mental model of the event [30,70]. By contrast, cause-and-effect relationships may be less clear-cut in people and the associated mental models. To illustrate, a person's behaviour (e.g., yelling at a stranger) can be explained by situational factors (e.g., because they were cut off in traffic) or by dispositional factors (e.g., because they have an aggressive personality), and each can provide a compelling causal explanation for the person's behaviour [73,74]. Given the greater variety of causal explanations, (mis)information in person mental models may be more malleable, and a retraction may be more easily accommodated [see 75]. Future research should directly contrast the factors that guide the CIE in event- and person-related reasoning by directly comparing event- and person-related misinformation within a single study—though this may present a nontrivial methodological challenge.

Misinformation coherence—the connection between the misinformation and other information—tends to be front and centre in event-related reasoning studies on the CIE, as well as in the broader belief perseverance literature [22,76]. However, including coherence-building elements in the studies reported here did not reliably yield a CIE in person impressions, suggesting misinformation coherence was not strong enough in this context. Given this, we might ask how misinformation coherence can be strengthened in person-related contexts. Person-related reasoning relies on social-category information about the protagonist (e.g., age, gender, occupation) and its inclusion may strengthen misinformation coherence [see 77,78]. When forming impressions of people who belong to a particular social group, we may rely on existing stereotypes or schemas to guide inferential reasoning [i.e., social identity theory, 79, see also 80–82]. Information that coheres with these prior beliefs may strengthen model coherence and contribute to belief perseverance. Support for this account comes from studies that have found convincing evidence of a CIE in person impressions [11,12,34]. In Thorson [12] the protagonist was a politician from an opposing political party, which may have activated a schema of distrust that cohered with the misinformation (i.e., the politician allegedly accepting a bribe). Similarly, in Mickelberg, Walker, Ecker, and Fay [11], the protagonist was a hospital patient with ambiguous symptoms which could be associated with a mental illness, which may have activated a schema [i.e., mental health stigma, see 83,84] that cohered with the misinformation (i.e., a schizophrenia or major depressive disorder diagnosis). By contrast, when forming an impression of a novel individual who is not tied to a particular group (as was done in the present study and others that have failed to find evidence for a CIE in person impressions), the absence of social-category information may have lowered misinformation coherence, and reduced the probability of finding a CIE in person impressions [9,10,13]. Future work could include social-category information about the protagonist such as race [e.g., 85], age [see 86], or gender [see 87], as this may activate pre-existing attitudes or stereotypes that strengthen misinformation coherence and moderate a CIE in person impressions.

## Conclusion

The negative impact of misinformation on a person's reputation is widely recognised, prompting research on whether misinformation can continue to influence person impressions following a retraction (the continued influence effect; CIE). The present study tested the impact of retracted misinformation on person impressions across three experiments. The experiments used different pieces of misinformation, but in each case the misinformation cohered with the other information provided about the protagonist. In each experiment, a single piece of negative misinformation had a strong negative impact on person impressions. However, once retracted there was mixed evidence of a CIE: one experiment showed a small continued influence effect, whereas the other two experiments showed no evidence for continued influence of retracted misinformation on person impressions. In sum, there was no conclusive evidence for a CIE in person impressions. Given the mixed findings, an internal meta-analysis was conducted. This also returned no conclusive evidence for a CIE in person impressions. To clarify the mixed results, future research should examine the factors that may moderate the emergence of a continued influence in person impressions, such as the specific attributes of the misinformation or the presence of social category information. This may reveal the mechanisms underlying the continued influence effect in person impressions, informing strategies to combat person-related misinformation.

## Supporting information

**S1 File. Pilot testing coherence-building statements.**
(PDF)

**S2 File. Impression formation task.**
(PDF)

**S3 File. Additional analyses.**
(PDF)

## Author contributions

**Conceptualization:** Amy J. Mickelberg, Bradley Walker, Ullrich K. H. Ecker, Piers D. L. Howe, Andrew Perfors, Nicolas Fay.

**Data curation:** Amy J. Mickelberg.

**Formal analysis:** Amy J. Mickelberg.

**Funding acquisition:** Amy J. Mickelberg, Nicolas Fay.

**Investigation:** Amy J. Mickelberg, Bradley Walker, Ullrich K. H. Ecker, Nicolas Fay.

**Methodology:** Amy J. Mickelberg, Bradley Walker, Ullrich K. H. Ecker, Nicolas Fay.

**Project administration:** Amy J. Mickelberg.

**Software:** Bradley Walker.

**Supervision:** Bradley Walker, Ullrich K. H. Ecker, Nicolas Fay.

**Visualization:** Amy J. Mickelberg.

**Writing – original draft:** Amy J. Mickelberg.

**Writing – review & editing:** Amy J. Mickelberg, Bradley Walker, Ullrich K. H. Ecker, Piers D. L. Howe, Nicolas Fay.

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
