## [Decision Letter · Decision Letter 0]

17 Dec 2024

PONE-D-24-44061Did he or didn’t he? Mixed evidence for the continued influence of retracted misinformation on person impressionsPLOS ONE

Dear Dr. Fay,

Thank you for submitting your manuscript to PLOS ONE. After careful consideration, we feel that it has merit but does not fully meet PLOS ONE’s publication criteria as it currently stands. Therefore, we invite you to submit a revised version of the manuscript that addresses the points raised during the review process. Please submit your revised manuscript by Jan 31 2025 11:59PM. If you will need more time than this to complete your revisions, please reply to this message or contact the journal office at plosone@plos.org . Please include the following items when submitting your revised manuscript:

We look forward to receiving your revised manuscript.

Kind regards,

Alessandra S. Souza, Ph.D.

Academic Editor

PLOS ONE

Journal Requirements:

 “This research was supported by a Postgraduate Research Scholarship from the Defence Science and Technology Group of the Department of Defence and an Australian Government Research Training Program Scholarship to the first author, an Australian Research Council grant FT190100708 to the third author, and an Office of National Intelligence and Australian Research Council grant NI210100224 to the last author.”

Additional Editor Comments:

I am sorry for the delay in sending the decision regarding this manuscript. As you know, it has become increasingly difficult to find reviewers. I had secured two reviewers to evaluate this submission, but the second reviewer never submitted their evaluation after several reminders. Given that it is almost holiday season (a period in which it becomes increasingly hard to find reviewers), I have decided to act based on the recommendation of one reviewer (comments appended below) and from my own reading. I am inviting a major revision, so that you have time to appreciate the comments and decide on the best course of action. Please consider all issues raised by Reviewer 1, and respond to it to the best of your abilities. I may invite a second reviewer if you decide to resubmit this manuscript for reevaluation in PLOS ONE. In addition to Reviewer 1's comments, I have a few comments derived from my own reading of the paper.(1) I did not understand the logic of the addition and the confirmation phrases. Why are they considered an addition or confirmation? I couldn't see any other phrase they were related to in the supplement. Please be more clear about the logic behind the methodology used.

(2) Regarding the data-analysis, you try to establish that there is no difference between the retraction and control conditions based on a non-significant p-value. This is not appropriate. Please include targeted analyses that can be used to establish that two conditions do not differ (e.g., Bayesian analysis; equivalence tests).(3) I was not convinced about the meta-analysis. I was under the impression at first that the meta-analysis included the articles previously published by the group (which seems to comprise most of the literature on this topic), but upon re-reading, I realized that it only included the 3 reported studies. Why not include all the available evidence on CIE with person impressions?

(4) As mentioned in the discussion, in Experiment 2, the average rating was close to 3 for the negative no-retraction group; in the other experiments, ratings were close to 4, which suggest people were more neutral. I wonder whether this indicates that the effect is moderated by the degree with which the misinformation was able to affect the formation of an impression. I believe more data is needed to address which conditions led to persistent effects on impressions. Perhaps including targets that are higher in emotionality (e.g., messing with the pension funds of the company; domestic violence; sexual harassment) could led to observations of persistent effects. This was not considered here as a potential moderator (this might help explain why in some prior studies a CIE was observed?), but it would be important in order to establish that retractions are more effective for person impressions. 

Reviewers' comments:

Reviewer's Responses to Questions

**Comments to the Author**

1. Is the manuscript technically sound, and do the data support the conclusions?

Reviewer #1: Partly

2. Has the statistical analysis been performed appropriately and rigorously? 

Reviewer #1: Yes

3. Have the authors made all data underlying the findings in their manuscript fully available?

Reviewer #1: Yes

4. Is the manuscript presented in an intelligible fashion and written in standard English?

Reviewer #1: Yes

5. Review Comments to the Author

Reviewer #1: I appreciate the opportunity to review the manuscript titled "Did he or didn’t he? Mixed evidence for the continued influence of retracted misinformation on person impressions" by Mickelberg et al. This study provides an interesting investigation into the continued influence effect (CIE) of retracted misinformation on person impressions. However, there are several areas where the manuscript can be improved to meet the standards of PLOS ONE.

1. Experimentation, Statistical Methods, and Technical Standards

P. 23: The paper mentions that the CIE was only detected in Experiment 2 and that the effect sizes were small. This discrepancy across experiments could be due to various factors, such as differences in the nature of the misinformation or the emotional content. I recommend expanding this section to address how different types of misinformation (e.g., moral vs. non-moral misinformation) might lead to stronger or weaker effects on person impressions.

P. 23: The authors suggest that dynamic vs. non-dynamic task designs may explain the differing results across experiments. However, to make a definitive claim, the authors should consider replicating the study with the same vignettes but varying the task design. This would offer more direct evidence for the impact of task design on the CIE. Without this, the suggestion remains speculative.

2. Testable Hypotheses and Scientific Contribution

P. 2: The hypotheses are clearly articulated, and the study addresses an important gap in the literature. However, the mixed results across the experiments raise questions about the robustness of the CIE. I recommend the authors provide a deeper analysis of these inconsistencies. The manuscript could benefit from more discussion on the potential reasons for these mixed results, including moral dimensions of the misinformation and counterfactual thinking.

P. 23: I also suggest the authors reference O'Rear & Radvansky (2020), which argues that the failure to accept retractions is related to the belief in the correction. This could provide additional theoretical context for explaining why the misinformation persisted in some experiments but not others. Additionally, exploring counterfactual thinking (how difficult it was for participants to generate alternatives to the misinformation) could help explain the variability in CIE across the experiments.

3. Appropriateness of the Design and Methods

P. 7: The coherence-building elements used in the experiments are mentioned, but the manuscript does not provide enough detail on how these elements were chosen. The authors should elaborate on the selection process. Specifically, the criteria that coherence-building statements were "unlikely to imply" but "likely to explain" the misinformation are not clearly explained (lines 239–240). This criterion should be better justified.

P. 23 (lines 542–544): The authors note that the misinformation in Experiment 2 involved an extreme act of violence (kicking a pet dog), which may have tapped into a different moral dimension compared to other misinformation (e.g., dishonesty). I suggest the authors explore the moral dimensions of misinformation more thoroughly and compare it to other studies.This could include comparisons to prior work like Ecker & Rodricks (2020) where vignettes seem to include violent behaviour (e.g. John slapped his girlfriend during an argument) (lines 493–504). If the moral dimension should be driving the CIE should this vignette not be more likely to show evidence of continued reliance on the retracted claim?

P. 19: The authors conduct an internal meta-analysis. However, it is unclear why the random-effects model was chosen over other models, and how this decision affects the generalizability of the findings. I recommend more discussion of the rationale behind choosing this model and its implications for interpreting the results.

4. Data Presentation and Conclusion

P. 26: The authors conclude that there is no conclusive evidence for the CIE in person impressions, which is somewhat understated in light of the mixed results. Given the mixed findings, the conclusion should engage more deeply with the broader literature on belief perseverance and misinformation correction. This would allow the authors to emphasise the implications of their findings, such as how misinformation can impact person impressions in everyday life or online interactions.

5. Writing and Clarity

The manuscript is generally well-written. However, the results section could benefit from clearer transitions between experiments and findings, especially in Experiment 2. The mixed results are not immediately clear, and it would help to clarify why Experiment 2 produced a different outcome compared to the other experiments.

6. Title, Abstract, and Figures

Abstract: The abstract could benefit from a more explicit mention of the mixed results observed across the three experiments. It would be helpful to briefly outline the implications of these mixed results for future research or real-world applications.

Figures: The figures are well-organised and support the data effectively. However, in Experiment 1, the error bars and bootstrapped confidence intervals are not sufficiently explained. I suggest including a more detailed description of these statistical details in the figure captions. For example, the number of resamples used in the bootstrapping process should be mentioned to enhance clarity.

7. References

The references are well-selected, but the authors should ensure consistency in citation formatting, particularly with the journal names and issue numbers. For example, entries such as "Ecker UKH, Lewandowsky S, Cook J, et al." should be reviewed for consistency with the PLOS ONE citation style guide.

Summary:

This is a promising paper that tackles an important topic in the field of the continued influence effect. However, given the mixed evidence across the three experiments, the paper is not yet ready for publication. To strengthen the manuscript, the authors should provide a more detailed analysis of the mixed results, particularly focusing on the moral dimensions of misinformation, task design, and potential alternative explanations, including the role of counterfactual thinking. If feasible, an additional study directly comparing the dynamic and non-dynamic task designs using the same vignette would provide more conclusive evidence regarding the impact of task design on the results. Finally, the manuscript would benefit from a thorough language edit to enhance clarity and address minor grammatical issues.

With these revisions, I believe the manuscript has the potential to make a valuable contribution to the field.

6. PLOS authors have the option to publish the peer review history of their article (what does this mean? ). If published, this will include your full peer review and any attached files.

**Do you want your identity to be public for this peer review?** For information about this choice, including consent withdrawal, please see our Privacy Policy .

Reviewer #1: No

---

## [Author Response · Author response to Decision Letter 0]

26 Feb 2025

Please see Response to PLOS ONE Reviewers document.

---

## [Decision Letter · Decision Letter 1]

16 Mar 2025

Did he or didn’t he? Mixed evidence for the continued influence of retracted misinformation on person impressions

PONE-D-24-44061R1

Dear Dr. Fay,

Thank you for submitting your manuscript to PLOS ONE. I have sent the paper back to the original reviewer in the first round and I have also read the paper to assess the implemented changes - as I have also made comments on the first submission. The reviewer was satisfied with the changes and so am I. R1 still made a few suggestions for changes in some setences, but they appear to me as more a matter of style, so I will live up to the authors if they want to incorporate these suggestions during the proof stage. As a minor note I would also suggest removing terms like "female" and "male" to refer to gender of the participants, as the new APA guidelines indicate that we should refer to "women" and "men" instead.

We’re pleased to inform you that your manuscript has been judged scientifically suitable for publication and will be formally accepted for publication once it meets all outstanding technical requirements.

Kind regards,

Alessandra S. Souza, Ph.D.

Academic Editor

PLOS ONE

Additional Editor Comments (optional):

Reviewers' comments:

Reviewer's Responses to Questions

**Comments to the Author**

1. If the authors have adequately addressed your comments raised in a previous round of review and you feel that this manuscript is now acceptable for publication, you may indicate that here to bypass the “Comments to the Author” section, enter your conflict of interest statement in the “Confidential to Editor” section, and submit your "Accept" recommendation.

Reviewer #1: All comments have been addressed

2. Is the manuscript technically sound, and do the data support the conclusions?

Reviewer #1: Yes

3. Has the statistical analysis been performed appropriately and rigorously? 

Reviewer #1: Yes

4. Have the authors made all data underlying the findings in their manuscript fully available?

Reviewer #1: Yes

5. Is the manuscript presented in an intelligible fashion and written in standard English?

Reviewer #1: Yes

6. Review Comments to the Author

Reviewer #1: Dear Authors,

Thank you for your revised manuscript and for addressing the comments from the first round of reviews. The revisions are generally well-implemented, and I appreciate the clarifications and additional analyses provided. That said, I have a few additional suggestions for improving the flow, clarity, and accuracy of certain sections.

P.27. “While investigation of misinformation reliance and its continued impact on person impressions is of some importance, studying this phenomenon may require greater statistical power than is usually attained in person-impression studies [56].” I think this sentence could be rewritten to allow for the premise to better support the rest of the sentence. Maybe something like: “Although studying the reliance on misinformation and its continued impact on person impressions is important, such research often requires greater statistical power than is typically achieved in person-impression studies.”

P.29. The categorization of "loyalty vs. disloyalty" in the context of violent behaviors, such as slapping a partner during an argument, seems somewhat problematic. While I understand that moral dimensions are crucial in categorizing behaviors, I would recommend reconsidering whether "loyalty vs. disloyalty" is the most appropriate moral dimension for this particular type of behavior. The issue of relationship violence might be better categorized under a different moral dimension, such as "relationship integrity" or "violence." I suggest revising this section to better reflect the complex nature of the moral dimensions at play here, and to avoid oversimplification.

Minor issues

"Reysen likability scale":

It should be "Reysen Likability Scale" (capitalizing "Likability" as it refers to the name of the specific scale).

"the retracted misinformation was fully discounted":

It could be clearer if written as: "the influence of retracted misinformation was fully discounted" or "the retracted misinformation had no effect." (abstract)

7. PLOS authors have the option to publish the peer review history of their article (what does this mean? ). If published, this will include your full peer review and any attached files.

**Do you want your identity to be public for this peer review?** For information about this choice, including consent withdrawal, please see our Privacy Policy .

Reviewer #1: No

---

## [Editor Report · Acceptance letter]

PONE-D-24-44061R1

PLOS ONE

Dear Dr. Fay,

I'm pleased to inform you that your manuscript has been deemed suitable for publication in PLOS ONE. Congratulations! Your manuscript is now being handed over to our production team.

Kind regards,

on behalf of

Dr. Alessandra S. Souza

Academic Editor

PLOS ONE